# Vision-Based Real-Time Bolt Loosening Detection by Identifying Anti-Loosening Lines

**DOI:** 10.3390/s24206747

**Published:** 2024-10-20

**Authors:** Wenyang Lei, Fang Yuan, Jiang Guo, Haoyang Wang, Zaiming Geng, Tao Wu, Haipeng Gong

**Affiliations:** 1School of Power and Mechanical Engineering, Wuhan University, Wuhan 430072, China; 2017302650054@whu.edu.cn (W.L.); 2023282080056@whu.edu.cn (H.W.); 2State Key Laboratory of Water Resources and Hydropower Engineering Science, Wuhan 430072, China; 3China Yangtze Power Co., Ltd., Yichang 443000, China

**Keywords:** bolt loosening detection, deep learning, computer vision, structural health monitoring

## Abstract

Bolt loosening detection is crucial for ensuring the safe operation of equipment. This paper presents a vision-based real-time detection method that identifies bolt loosening by recognizing anti-loosening line markers at bolt connections. The method employs the YOLOv10-S deep learning model for high-precision, real-time bolt detection, followed by a two-step Fast-SCNN image segmentation technique. This approach effectively isolates the bolt and nut regions, enabling accurate extraction of the anti-loosening line markers. Key intersection points are calculated using ellipse and line fitting techniques, and the loosening angle is determined through spatial projection transformation. The experimental results demonstrate that, for high-resolution images of 2048 × 1024 pixels, the proposed method achieves an average angle detection error of 1.145° with a detection speed of 32 FPS. Compared to traditional methods and other vision-based approaches, this method offers non-contact measurement, real-time detection capabilities, reduced detection error, and general adaptability to various bolt types and configurations, indicating significant application potential.

## 1. Introduction

Bolts are widely used as fasteners in industries such as automotive, aerospace, construction, and manufacturing, where they are essential for maintaining structural integrity and operational efficiency by securely joining components [1,2]. However, bolted joints can be compromised by factors like environmental conditions, dynamic loads, material degradation, and improper installation. Bolt loosening or failure poses significant risks to equipment performance, product quality, and, most critically, personnel safety [3]. As a result, the timely detection and prevention of bolt loosening is a key concern across various industries.

Early methods for detecting bolt loosening primarily relied on manual inspection techniques, such as anti-loosening wires, magnetic suction, and torque measurement. The accuracy of these methods heavily depended on the experience and expertise of the inspectors, leading to a significant workload and limited detection precision [4]. With advancements in structural health monitoring, contact sensor-based methods for bolt loosening detection have gained prominence. These include techniques utilizing piezoelectric impedance [5,6,7], acoustic waves [8,9], and vibration response [10,11], which involve installing sensors near the bolts and analyzing the resulting signal characteristics to assess loosening. However, these sensor-based approaches are often susceptible to environmental influences like temperature, humidity, and electromagnetic interference. Additionally, the extensive use of sensors can escalate monitoring costs, and the need to interpret complex relationships between sensor data and bolt conditions increases the complexity of data processing [12].

Significant advancements have been made in detecting bolt loosening using contact sensor signals through advanced deep learning methods, which can accurately map sensor data to bolt loosening conditions [11]. For instance, Kong et al. [13] introduced an impact-based non-destructive technique that assesses bolt health by examining acoustic changes when loosening occurs. This approach utilizes power spectral density as the signal feature and applies a decision tree machine learning model to classify the data generated by impacts, effectively identifying the bolt’s looseness. Similarly, Gao et al. [14] developed a nondestructive method based on ultrasonic echoes to detect looseness in blind hole bolts. Their method transforms ultrasonic echo signals into image representations using wavelet analysis and employs a convolutional neural network (CNN) to classify and predict these images, thus determining the bolt’s looseness. Although these methods provide high accuracy, robustness against noise, and effective feature extraction, they are highly dependent on large amounts of data and the extensive use of sensors for signal acquisition. Furthermore, in challenging environments, external factors can introduce interference, which may compromise the consistency and reliability of the detection results.

With advancements in computer vision and deep learning technologies, non-contact visual inspection methods have garnered attention due to their low cost, ease of deployment, and non-intrusive measurement capabilities [15]. These methods can be broadly categorized into two types based on detection criteria. The first type focuses on variations in the protrusion length of bolt heads or studs. For instance, Cha et al. [16] used the Hough Transform to calculate the horizontal and vertical lengths of bolt heads from images and classified the degree of looseness using a support vector machine (SVM). Similarly, Ramana et al. [17] employed the Viola–Jones algorithm combined with an SVM to locate and identify loose bolts. Gong et al. [18] proposed a method that utilizes deep learning and geometric imaging principles to quantitatively calculate the protrusion length of bolts, thereby identifying bolt looseness. The second type determines bolt looseness by calculating the relative rotation angle between different parts of the bolt. Sohn et al. [19] developed a detection method based on image feature tracking, identifying looseness by rotating features in bolt images. Huynh and Thanh-Canh [20] designed a method integrating Fast R-CNN for bolt detection, image correction, angle calculation, and looseness determination, achieving a 93% accuracy rate in practical tests. Qi et al. [21] combined Fast R-CNN with Hough Transform to detect the loosening angle of bolts via marked lines. Deng et al. [22] utilized Keypoint R-CNN and geometric imaging principles to calculate the loosening angle by identifying key points on the broken loosening lines on the bolts. While the first type of method typically offers lower identification accuracy, the second type often relies on traditional image processing techniques that lack precision and robustness. In practical applications, these methods often depend on the historical state of the bolts, and their detection accuracy is influenced by the camera’s shooting angle. Additionally, there are significant limitations regarding the real-time detection and general applicability of these methods.

This paper introduces a generalized real-time vision-based method for detecting bolt looseness by analyzing anti-loosening marks on the bolt. The method begins with high-precision, real-time bolt detection using You Only Look Once version 10 (YOLOv10-S). Next, a two-step fast segmentation convolutional neural network (Fast-SCNN) is employed for image segmentation: the first step extracts the bolt stud and nut top surfaces, while the second step isolates the anti-loosening marks. Based on these segmentation results, ellipse and line fitting algorithms are used to calculate key intersection points between the anti-loosening marks and the bolt connection. Finally, spatial projection relationships are utilized to determine the actual corresponding points and compute the bolt’s looseness angle. The key advantages of this method are as follows: (1) it enables non-contact detection of bolt looseness without the need for historical data on the bolt’s condition; (2) it ensures real-time detection through the integration of YOLOv10-S, Fast-SCNN, ODR ellipse fitting, and TLS line fitting algorithms, maintaining high precision and efficiency; and (3) it offers broad applicability across different types of bolts by utilizing anti-loosening mark recognition, making it adaptable to a wide range of practical scenarios.

## 2. Methodology

### 2.1. Overview

The bolt anti-loosening line is a commonly used marking for detecting bolt looseness. Initially, a marking line is drawn on the bolt and its adjacent area when it is fully tightened, as shown in Figure 1. Upon loosening, this marking line on the nut and stud sections breaks at the junction circle. The extent of this breakage correlates with the degree of bolt looseness, indicating the looseness angle.

This paper presents a universal real-time method for detecting bolt looseness using deep learning and computer vision techniques. The method determines if a bolt is loose by calculating the relative rotation angle between the nut and the stud through the detection of anti-loosening line markers on the bolt. As illustrated in Figure 2, the process begins with employing YOLOv10-S to efficiently detect bolts in images and extract their regions. Next, Fast-SCNN is utilized for two-step image segmentation: first, segmenting the stud side and nut top surfaces, and then extracting the anti-loosening line markers from both the stud and nut. Subsequently, the intersection points between the projected ellipse at the nut–stud junction and the detected anti-loosening lines are calculated based on the segmentation results. Finally, spatial projection transformations are applied to compute the bolt’s looseness angle from these identified intersection points.

### 2.2. Bolt Detection with YOLOv10-S

YOLOv10 represents the latest advancement in the YOLO series, emphasizing both efficiency and accuracy. It features a lightweight classification head, spatial–channel decoupled downsampling (SCDown), and a rank-based block design to minimize computational redundancy and enhance overall efficiency. Additionally, YOLOv10 incorporates large kernel convolutions and partial self-attention (PSA) modules to bolster model performance. The inclusion of SCDown enables the network to decouple spatial and channel information during the downsampling process, which reduces computation while preserving important features. The SPPF (spatial pyramid pooling fast) further strengthens the model’s ability to detect objects at multiple scales, which is particularly beneficial in scenarios involving objects of different sizes and orientations, such as bolts. YOLOv10-S is designed with dual-label assignment and consistent matching metrics that enable NMS-free postprocessing. This approach eliminates the need for traditional non-maximum suppression (NMS) by ensuring that the model generates high-confidence, accurate predictions directly during inference. The result is faster and more efficient predictions without sacrificing precision. With the inclusion of partial self-attention (PSA), the model can more effectively focus on key regions in an image while filtering out irrelevant background noise, improving overall detection accuracy in complex environments [23]. For this study, YOLOv10-S has been chosen for bolt detection. The structure of YOLOv10-S is illustrated in Figure 3.

### 2.3. Two-Step Segmentation with Fast-SCNN

To accurately extract anti-loosening line markers, which are partially located on both the stud and the nut, precise segmentation of these areas is essential. This segmentation reduces noise and enables accurate marker extraction. The process is enhanced by employing a two-step segmentation method, as shown in Figure 4.

Fast-SCNN, introduced by Google in 2019, is a lightweight, real-time image segmentation model designed to balance computational efficiency and segmentation performance. Its architecture features a two-branch encoder–decoder structure with skip connections, effectively capturing both local and global features [24]. Due to these advantages, Fast-SCNN is chosen for image segmentation in this study.

As illustrated in Figure 5, Fast-SCNN consists of four key components: a learning-to-downsample module, a global feature extractor, a feature fusion module, and a classifier. The learning-to-downsample module rapidly reduces input size, while the global feature extractor captures coarse, high-level features. The feature fusion module integrates these features to create a comprehensive representation, which the classifier then uses to generate final segmentation maps. Fast-SCNN’s design ensures efficient performance on resource-constrained devices while maintaining high segmentation accuracy.

### 2.4. Key Intersections Acquisition

Based on the image segmentation results from the previous step, the junction line between the stud and the nut can be identified. This line represents a partial elliptical curve, which is the projection of the junction circle between the nut and the stud onto the camera plane. As shown in Figure 6, to accurately calculate the loosening angle, it is essential to determine the intersection points between the two anti-loosening markers and this junction ellipse. Using spatial projection transformation, the corresponding points on the actual junction circle of the stud and nut are then obtained. This section outlines the process of calculating these two critical intersection points through ellipse fitting and line fitting.

#### 2.4.1. Ellipse Fitting

Common ellipse fitting methods include ordinary least squares (OLS), total least squares (TLS), least squares median (LSM), orthogonal distance fitting (ODF), and RANSAC. Since the elliptical contours obtained from the image segmentation algorithm generally lack distinct outliers but may contain noise, it is essential to ensure real-time fitting and robustness against noise [25]. After evaluating the fitting accuracy and speed of various methods, orthogonal distance regression (ODR) was selected for ellipse fitting. The ODR method applied for ellipse fitting is outlined in Algorithm 1.

The ODR ellipse fitting method minimizes the perpendicular distances from data points to the fitted model, accounting for errors in both independent and dependent variables. This approach is particularly effective when noise affects both variables, providing a more accurate and robust fit. ODR excels in stability and precision, especially in the presence of measurement errors, making it ideal for applications requiring high noise tolerance.
**Algorithm 1.** Orthogonal distance regression for ellipse fitting.**Input:** Data points $\left(x_{i},\;y_{i}\right)$ of ellipse contours, initial parameters = (($c_{1\_0}$, $c_{2\_0}$), $a_{0}$, $b_{0}$, $\theta_{0}$), max iteration, tolerance.**Output:** Ellipse parameters (($c_{1}$, $c_{2}$), a, b, θ)**1:** *p* = initial parameters**2: For** iteration = 1 to max iteration **do**:**3:**   **For** each ($x_{i},\;y_{i}$) **in** data points **do:****4:**    $d_{i}$ = orthogonal distance from point ($x_{i},\;y_{i}$) to the ellipse defined by parameters *p***5:**    $r_{i}\;$= residual of the ellipse equation**6:**   **End For****7:**   $F\left(p\right)=\sum \left({d_{i}}^{2}+{r_{i}}^{2}\right)$ **For** all i**8:**   J = Jacobian matrix of the objective function’s partial derivatives with respect to each parameter in p**9:**   $p_{update}\;$= calculate the parameter update step using Levenberg–Marquardt**10:**   $p_{new}=p-p_{update}$**11:**   **If** $\max\left(\left|p_{new}-p\right|\right)<tolerance$ then:**12:**    **Break loop****13: End For****14: Return** optimized *p* (($c_{1}$, $c_{2}$), a, b, θ)

#### 2.4.2. Linear Fitting

Common linear fitting methods include ordinary least squares (OLS), total least squares (TLS), Theil–Sen estimator, and RANSAC. Like orthogonal distance regression, TLS minimizes perpendicular distances from data points to the fitting curve, making it effective for cases where errors affect both independent and dependent variables. TLS is particularly suitable for fitting lines with significant width and noise, including vertical lines [26]. Given these characteristics, TLS is selected for fitting anti-loosening marks. The procedure for straight line fitting using ODR is detailed in Algorithm 2.
**Algorithm 2.** Total least squares for line fitting.**Input:** Data points (*x*_1_, *y*_1_), (*x*_2_, *y*_2_), …, (*x*_n_, *y*_n_) of the marker line;**Output:** Line equation: ax+by+c=0, where (a, b, c) are the line coefficients;**1:** $x_{mean}\;$ = mean($x_{i}$ **for** *i* = 1 to *n*) **2:** $y_{mean}\;$ = mean($y_{i}$ **for** *i* = 1 to *n*)**3:** $x_{centered}$ = [$x_{i}-x_{mean}$ **for** *i* = 1 to *n*]**4:** $y_{centered}$ = [$y_{i}-y_{mean}$ **for** *i* = 1 to *n*]**5:** Matrix A = [[$x_{centered}\left[i\right]$, [$y_{centered}\left[i\right]$] for *i* = 1 to *n*]**6:** Compute covariance matrix $C=A^{T}A$**7:** Perform singular value decomposition (SVD) on the covariance matrix: $\left[U,\;S,\;V^{T}\right]=SVD(C)$**8:** Eigenvector (a, b) corresponds to the line equation ax+by+c=0**9:** $c=-(ax_{mean}+by_{mean})$**11:** Normalize (a, b, c)**12: Return** the line coefficients (a, b, c)

#### 2.4.3. Intersections Calculation

After obtaining the elliptic curve and the two anti-loosening marking straight lines, we can calculate the two intersections between the two straight lines and the elliptic curve in the image display part, i.e., the key intersection points that we need to request. The calculation process is shown in Algorithm 3.
**Algorithm 3.** Intersection calculation.**Input:** Ellipse, Line1, Line2, points of marking Line1, points of marking Line2**Output:** Intersection1, Intersection2**1:** Calculate intersections ($P_{1}$, $P_{2}$) of Ellipse with Line1**2:** Calculate intersections ($P_{3}$, $P_{4}$) of Ellipse with Line2**3:** Compute midpoint $m_{1}$ from points of Line1**4:** Compute midpoint $m_{2}$ from points of Line2**5:** Determine Intersection1 as the point which is closer to $m_{1}$ from ($P_{1}$, $P_{2}$)**6:** Determine Intersection2 as the point which is closer to $m_{2}$ from ($P_{3}$, $P_{4}$)**7: Return** Intersection1, Intersection2

### 2.5. Loosening Angle Calculation

The two key intersection points represent the relative rotation of the anti-loosening marks on the junction circle between the nut and the stud, as illustrated in Figure 7. By referencing the elliptic curve obtained in the previous step, the corresponding junction circle can be determined. The angle of bolt loosening is then calculated by determining the positions of these two key intersection points on the circle and measuring the angle between them.

According to the ellipse fitting result, we can establish that the center of the ellipse is *E*(*x*_0_, *y*_0_, 0), the long axis is 2*a*, the short axis is 2*b*, and the angle between the long axis and the *x*-axis is *θ*. As presented in Figure 8, assuming that the center of the circle corresponding to the ellipse of the stud–nut junction is *C*, and the height of the center of the circle in the *z*-axis is *z*_0_, then the center of the circle is the coordinates of *C*(*x*_0_, *y*_0_, *z*_0_), and the radius of the circle is *a*; then, the points on the circle *P*(*x*, *y*, *z*) should be satisfied:
(1)$$(x-x_{0}{)}^{2}+(y-\;y_{0}{)}^{2}+(z-z_{0}{)}^{2}=a^{2}$$

Suppose the vertices on the short and long axes on the ellipse are *A* and *B*, respectively; then, the coordinates of *A* and *B* can be obtained as ($x_{0}+b\sin\theta$, $y_{0}+b\cos\theta$, 0), ($x_{0}-a\cos\theta$, $x_{0}+a\sin\theta$, 0). Then, the points *A*′, *B*′ on the circles corresponding to *A* and *B* are, respectively, *A*′ ($x_{0}+b\sin\theta$, $y_{0}+b\cos\theta$, $z_{0}-\sqrt{a^{2}-b^{2}}$) and *B*′ ($x_{0}-a\cos\theta$, $x_{0}+a\sin\theta$, $z_{0}$). Then,
(2)$$\vec{\mathrm{CA}}=(\begin{array}{c} \mathrm{bsin}\theta ,\mathrm{bcos}\theta ,-\sqrt{a^{2}-b^{2}}) \end{array}\;\vec{\mathrm{CB}}=(-\mathrm{acos}\theta ,\;\mathrm{asin}\theta ,\;0)\;\vec{\mathrm{CP}}=(x-x_{0},\;y-y_{0},\;z-z_{0})$$

Clearly, $\vec{\mathrm{CA}}$, $\vec{\mathrm{CB}}$, $\vec{\mathrm{CP}}$ are coplanar; then, there exist
(3)$$\left|\begin{array}{ccc} x-x_{0} & y-y_{0} & z-z_{0}\\ -\mathrm{acos}\theta & \mathrm{asin}\theta & 0\\ \mathrm{bsin}\theta & \mathrm{bcos}\theta & -\sqrt{a^{2}-b^{2}} \end{array}\right|=0$$

Namely,
(4)$$\sqrt{a^{2}-b^{2}}\sin\theta \left(x-x_{0}\right)+\sqrt{a^{2}-b^{2}}\cos\theta \left(y-y_{0}\right)+b\left(z-z_{0}\right)=0$$

Then, the circle corresponding to the ellipse is
(5)$$\left\{\begin{array}{c} (x-x_{0}{)}^{2}+(y-y_{0}{)}^{2}+(z-z_{0}{)}^{2}=a^{2}\\ \sqrt{a^{2}-b^{2}}\sin\theta \left(x-x_{0}\right)+\sqrt{a^{2}-b^{2}}\cos\theta \left(y-y_{0}\right)+b\left(z-z_{0}\right)=0 \end{array}\right.$$

According to the solution of the key focus in the previous step, the coordinates of the two key points are $K_{1}(x_{1},\;y_{1},\;0)$ and $K_{2}(x_{2},\;y_{2},\;0)$. According to the equation of the circle and the correspondence of the points, their corresponding points on the circle are
$${K_{1}}^{\prime }(x_{1},y_{1},z_{1})$$
$${K_{2}}^{\prime }(x_{2},y_{2},z_{2})$$

The angle corresponding to the arc of the circle between these two points is
(6)$$\theta =\cos^{-1}\frac{\overset{\cdots \;\cdots \;\cdots \;\to }{C{K_{1}}^{\prime }}\cdot \overset{\cdots \;\cdots \;\cdots \;\to }{C{K_{2}}^{\prime }}}{\left|\overset{\cdots \;\cdots \;\cdots \;\to }{C{K_{1}}^{\prime }}|\;|\overset{\cdots \;\cdots \;\cdots \;\to }{C{K_{2}}^{\prime }}\right|}$$

## 3. Experiment

### 3.1. Data Acquisition

To preliminarily assess the feasibility of the proposed method, we conducted experimental validation on a simplified platform with 36 bolt connections. The setup included two types of bolts: 12 with round nuts and 24 with hexagonal nuts, all with an M16 inner diameter. These bolts were securely fastened to a metal plate, with anti-loosening marks applied when fully tightened. The bolts were then randomly loosened at angles of 5°, 10°, 30°, 60°, 90°, 120°, and 150°, with photographs taken after each adjustment. During each capture, the camera angle was maintained between 30° and 50° relative to the vertical axis, ensuring clear visibility of the anti-loosening marks in the horizontal direction. An industrial high-speed camera with a resolution of 2048 × 1024 was used to capture images of the platform, resulting in a total of 80 images for subsequent analysis. The experimental setup is depicted in Figure 9, Figure 10 and Figure 11.

### 3.2. Model Training

The model training consists of two main parts. First, YOLOv10-S is trained for bolt target detection. Second, three Fast-SCNN models are trained: the first for segmenting the stud side and nut top surface, the second for extracting markings from the stud side, and the third for extracting markings from the nut surface. The experimental setup for model training and validation is shown in Table 1.

To accurately evaluate the performance of the trained models, we used precision, recall, F1 score, and FPS to assess YOLOv10-S. Precision measures the proportion of true positive detections among all positive detections, indicating the model’s prediction accuracy. Recall measures the proportion of true positive instances correctly identified by the model, reflecting its ability to capture all relevant instances. F1 score is the harmonic mean of precision and recall, providing a single metric that balances both precision and recall. And FPS measures the model’s processing speed for images. Fast-SCNN was evaluated using intersection over union (IoU) and FPS. IoU quantifies the overlap between the predicted bounding box and the ground truth bounding box, defined as the ratio of the intersected area to the combined area. A higher IoU value indicates more accurate object localization. FPS similarly measures the speed of Fast-SCNN’s image segmentation. Table 2 presents the confusion matrix, and the formulas for calculating precision, recall, IoU, and FPS are as follows.
(7)$$\mathrm{FPS}=\frac{1000}{\begin{array}{c} \mathrm{Average}\;\mathrm{time}\;\mathrm{to}\;\mathrm{process}\\ \mathrm{an}\;\mathrm{image} \end{array}}$$
(8)$$\mathrm{Precision}=\frac{\mathrm{TP}}{\mathrm{TP}+\mathrm{FP}}$$
(9)$$\mathrm{Recall}=\frac{\mathrm{TP}}{\mathrm{TP}+\mathrm{FN}}$$
(10)$$F1=\frac{2*(\mathrm{Precision}*\mathrm{Recall})}{\left(\mathrm{Precision}+\mathrm{Recall}\right)}$$
(11)$$\mathrm{IoU}=\frac{\mathrm{Area}\mathrm{of}\mathrm{Intersection}}{\mathrm{Area}\mathrm{of}\mathrm{Union}}$$

#### 3.2.1. YOLOv10-S Training

The collected images were annotated and divided into a training set (80%) and a validation set (20%), forming the image dataset used for training the YOLOv10-S model. Then, we finetuned the pretrained YOLOv10-S, using 500 training epochs, a batch size of 64, and an initial learning rate of 0.001, with other parameters set to their default values. The loss function used for model training is composed of three components:(12)$$\mathrm{loss}\;={\mathrm{loss}}_{\mathrm{box}}+{\mathrm{loss}}_{\mathrm{cls}}+{\mathrm{loss}}_{\mathrm{dfl}}$$
where ${\mathrm{loss}}_{\mathrm{box}}$ is bounding box regression loss, ${\mathrm{loss}}_{\mathrm{cls}}$ is class prediction loss, and ${\mathrm{loss}}_{\mathrm{dfl}}$ is distribution focal loss. When training reached 400 epochs, the model accuracy was no longer improved and training was stopped. The loss, precision, and recall are illustrated in Figure 12.

After training, YOLOv10-S achieved a detection accuracy of 99.3% and a recall of 99.75% on the test set. The model was tested on 100 images, with an average inference time of 10.3 ms per image, resulting in a speed of 97 FPS, which meets the real-time requirements of practical detection scenarios. One test example is shown in Figure 13.

#### 3.2.2. Fast-SCNN Training

Bolts are detected and isolated from the images using YOLOv10-S, resulting in individual bolt images. These images are then segmented to label the stud sides and nut top areas, creating an image dataset of 167 segmented images. Of these, 80% were divided into a training set and 20% into a validation set. Fast-SCNN is employed for the initial training phase, with the following parameters: an initial learning rate of 0.001, a batch size of 32, a weight decay of 0.0001, and 500 training epochs. All other training parameters are kept at their default settings and the model is trained using cross-entropy loss.

After completing the first phase of training, the model was applied to bolt images, producing separate images of the stud side and the nut top. These images were then labeled to create two distinct image segmentation datasets, and 80% of them were divided into a training set and 20% into a validation set. We used the same training parameters as in the first phase to train two additional models, one for extracting the anti-loosening markers from the stud side and the other from the nut top.

The changes in the loss function and mIoU during both training phases are illustrated in Figure 14 and Figure 15. The first model achieved an accuracy of 97.54% in the first phase. In the second phase, the two models for marker extraction reached an mIoU of 95.86% and 96.21%, respectively. We also tested the processing speed of both steps, with an average image segmentation time of 10 ms per image, resulting in a detection speed of 50 FPS on the two steps. An example of image segmentation using Fast-SCNN is as Figure 16.

### 3.3. Key Intersections Calculation

#### 3.3.1. Comparison of Ellipse Fitting Methods

We compared several ellipse fitting methods, including ordinary least squares, total least squares, least squares median, orthogonal distance fitting, and RANSAC. Using a normally distributed set of 100 randomly generated ellipses, each with 60 noisy points, we evaluated the fitting accuracy and speed of these methods. The results are summarized in Table 3.

Among the ellipse fitting methods compared, orthogonal distance regression exhibited superior performance in both fitting speed and accuracy. An example of ellipse contour fitting using ODR is shown in Figure 17.

#### 3.3.2. Comparison of Line Fitting Methods

We compared several line fitting methods, including ordinary least squares, total least squares, Theil–Sen estimator, and RANSAC. Fifty sets of lines were randomly generated using a normal distribution, with 100 noisy points created for each line. These methods were then applied to fit the lines, and their average fitting error and speed were calculated. The results are presented in Table 4. Among all the fitting methods compared, TLS demonstrated superior performance in both fitting speed and accuracy. The result of fitting the extracted marking lines using TLS is shown in Figure 18.

#### 3.3.3. Calculation of Key Intersections

After fitting the ellipse and anti-loosening marks, the two key intersections were calculated according to the logic of Algorithm 3. The average computation time for this process, over 100 iterations, was 0.02 ms, meeting the requirements for real-time processing. The calculated key intersections are shown in Figure 19.

### 3.4. Loosening Angle Calculation

The bolt loosening angles were then calculated using the coordinates of the key intersection points and the ellipse equation, based on the proposed angle estimation method. The calculation errors for different loosening angles are summarized in Table 5, with an average error of 1.145° across all bolts, indicating high accuracy in angle estimation. The main sources of error in angle estimation include variations in image acquisition conditions, such as camera angle, distance, and lighting, which may affect the clarity and visibility of the anti-loosening marks. Additionally, there is uncertainty in measuring the actual angle between the marks due to the width of the lines and slight irregularities in their drawing. The image segmentation algorithm (Fast-SCNN) may introduce pixel-level errors when extracting the marks, impacting the accuracy of subsequent line fitting. Although the total least squares (TLS) method was used for line fitting, it is still susceptible to errors when handling noisy data. Finally, surface irregularities on the bolt may cause slight distortion in the captured ellipse, affecting the accuracy of ellipse fitting. All these reasons contribute to the error between the loosening angle calculated by the algorithm and the measured loosening angle.

Further analysis was conducted on the angle calculation results for round and hexagonal nuts separately, and the result is summarized in Table 6. It was observed that the error for round nuts was significantly lower than that for hexagonal nuts. This discrepancy, as illustrated in Figure 20, is attributed to the greater sidewall thickness of the round nuts compared to the hexagonal nuts. The increased thickness results in higher-quality markings on the round nuts, leading to more precise extraction of the anti-loosening lines.

The complete method was applied to the collected images, with some detection examples shown in the accompanying Figure 21. The method accurately identifies the bolts and determines their loosening angles. For images with a resolution of 2048 × 1024, the average detection time per image was 40.2 ms. Specifically, YOLOv10-S took 10.3 ms for object detection, the two-step Fast-SCNN process took 21 ms, angle calculation took 2.6 ms, and the remaining operations took 1.3 ms. This results in a detection speed of 32 FPS, which meets the requirements for real-time detection.

The result revealed that the lower detection error for round nuts is primarily due to their smoother and thicker sidewall structure, which allows for clearer and more consistent anti-loosening line markings during image segmentation. In contrast, the thinner sidewalls of hexagonal nuts result in shorter markings that are more prone to being affected by light and shadows, leading to blurred segmentation and increased angle calculation errors. This effect is particularly evident under complex lighting conditions. Additionally, while the YOLOv10-S and Fast-SCNN processes account for the majority of the processing time, their lightweight design and fast performance ensure that the overall detection speed remains at 32 FPS, which is crucial for real-time monitoring, especially with high-resolution images. The experiments also demonstrated the model’s robustness in handling different types of bolts and loosening angles. The error distribution across various bolt structures indicates that the method is not only suitable for a wide range of bolts but also maintains stable accuracy and speed in complex real-world scenarios, highlighting its practical application potential.

## 4. Conclusions

In this study, we proposed a real-time vision-based method for detecting bolt loosening by identifying anti-loosening lines on bolt connections. By leveraging the latest advancements in computer vision and deep learning, particularly the YOLOv10-S and Fast-SCNN models, our method achieves high accuracy and speed in identifying bolt loosening and anti-loosening mark extraction. Furthermore, the combination of ellipse fitting and line fitting algorithms allows for accurate calculation of the bolt’s loosening angle by identifying key intersections on the nut–stud interface.

The experimental results demonstrated that our method can robustly detect bolt loosening at various angles, providing reliable performance under challenging imaging conditions. The proposed approach offers significant advantages over traditional methods, including non-contact measurement, real-time detection capabilities, and general applicability to different bolt types and configurations. The main contributions of this study are as follows:

(1) A real-time bolt loosening detection method based on computer vision is proposed, integrating the YOLOv10-S deep learning model with Fast-SCNN image segmentation techniques. This method enables direct recognition of anti-loosening line markers at bolt connections without relying on the historical state of the bolts.

(2) The proposed method achieves an average angle detection error of 1.145° and a detection speed of 32 FPS under high-resolution images of 2048 × 1024 pixels. It significantly outperforms traditional detection methods and other vision-based approaches, striking a favorable balance between detection accuracy and speed.

(3) The method demonstrates good adaptability, effectively handling different types of bolts, and is applicable to various complex monitoring scenarios, showcasing extensive practical application potential. 

This method has the potential to enhance the safety and reliability of bolted joints in various industrial applications by enabling timely detection and intervention for bolt loosening. Future work will focus on further refining the method, exploring its scalability, and testing its performance in more complex real-world scenarios.

## Figures and Tables

**Figure 1 sensors-24-06747-f001:**
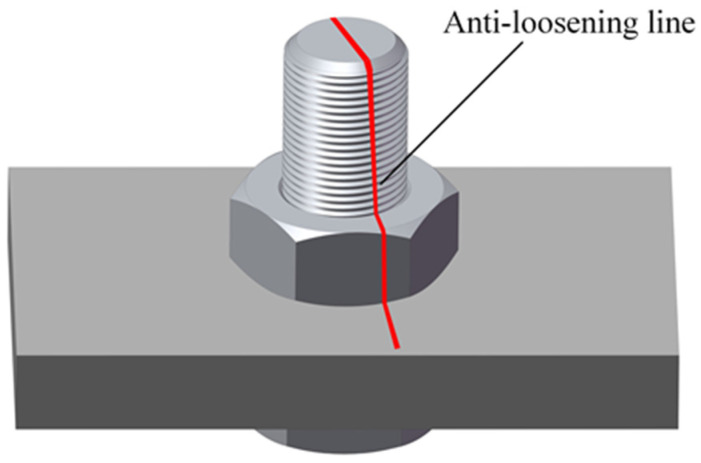
Anti-loosening mark.

**Figure 2 sensors-24-06747-f002:**
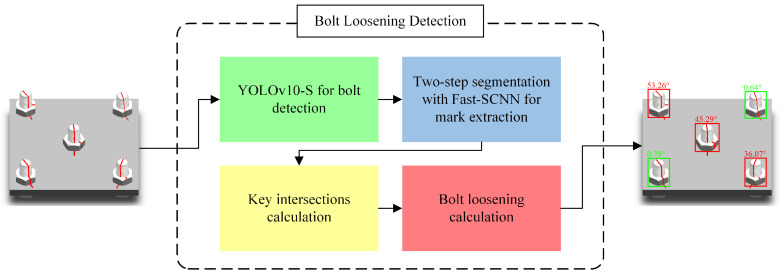
Overall methodological flow.

**Figure 3 sensors-24-06747-f003:**
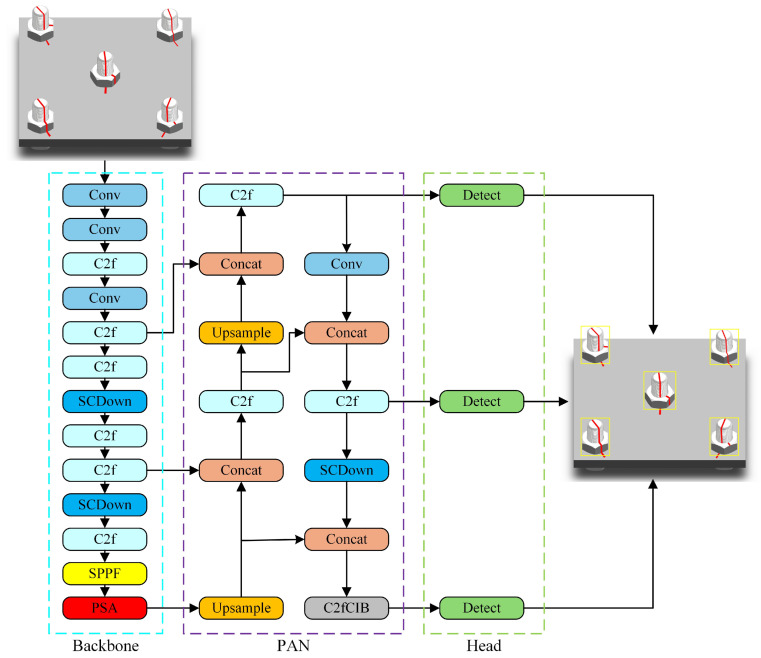
Structure of YOLOv10.

**Figure 4 sensors-24-06747-f004:**
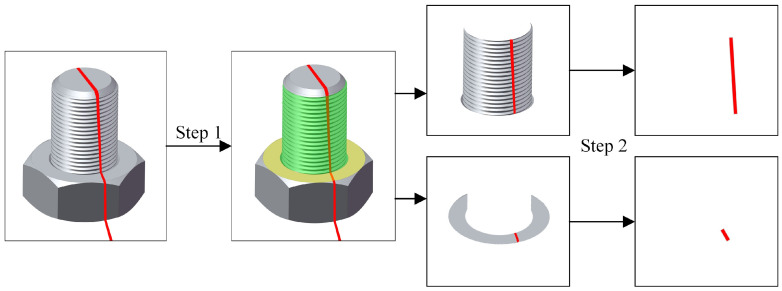
Two-step image segmentation to extract anti-loosening markers.

**Figure 5 sensors-24-06747-f005:**
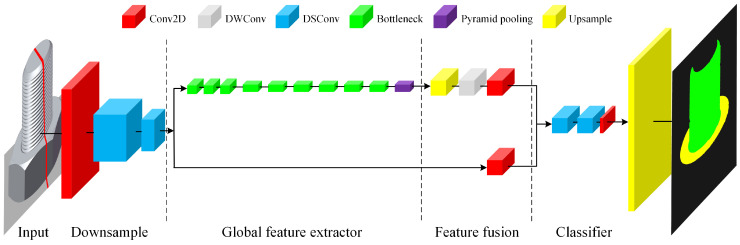
Structure of Fast-SCNN.

**Figure 6 sensors-24-06747-f006:**
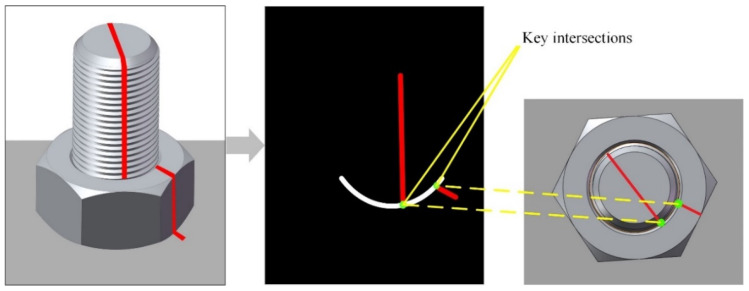
Location of key intersections.

**Figure 7 sensors-24-06747-f007:**
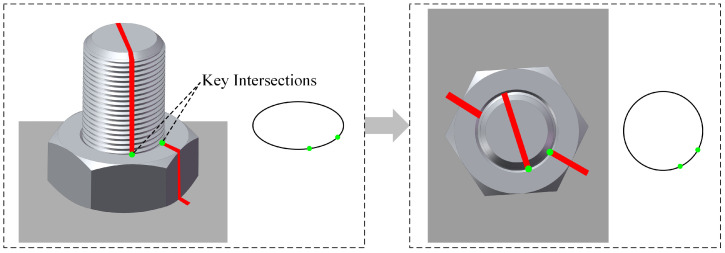
Correspondence of key intersections.

**Figure 8 sensors-24-06747-f008:**
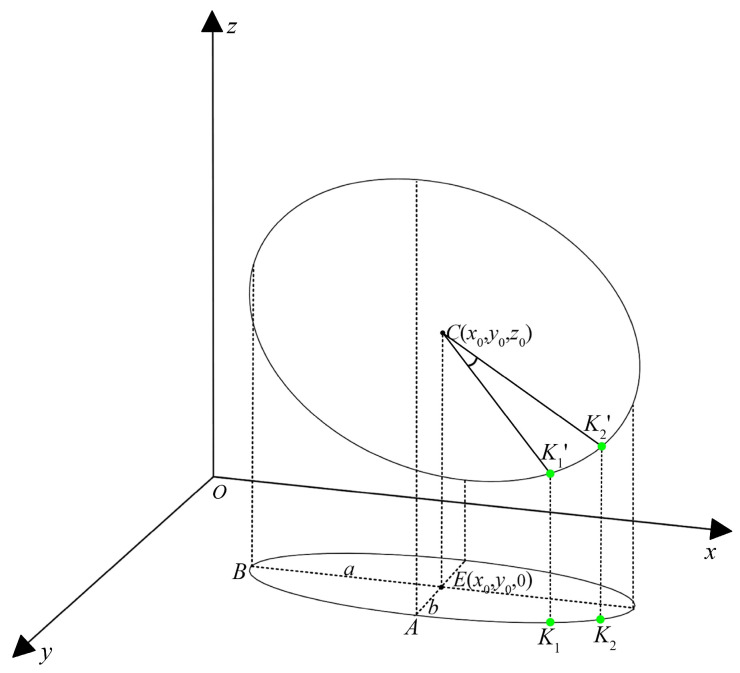
Projection relationships of intersecting circles projected into ellipses.

**Figure 9 sensors-24-06747-f009:**
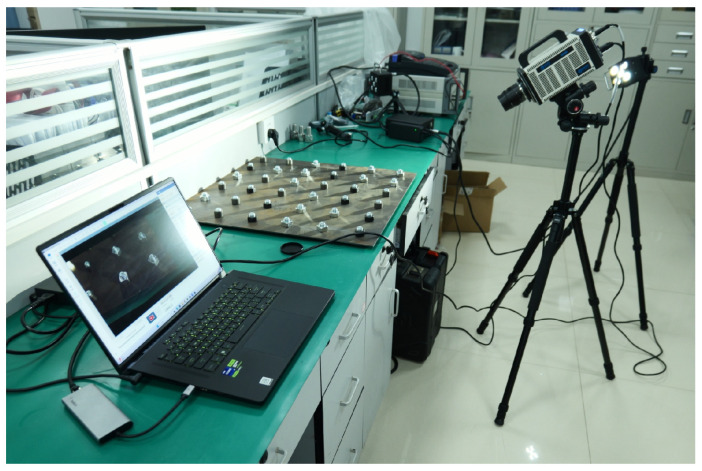
Experimental platforms.

**Figure 10 sensors-24-06747-f010:**
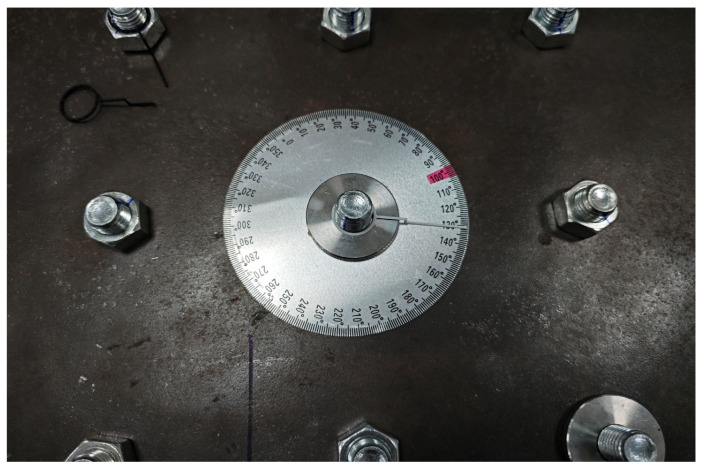
Loose angle measurement.

**Figure 11 sensors-24-06747-f011:**
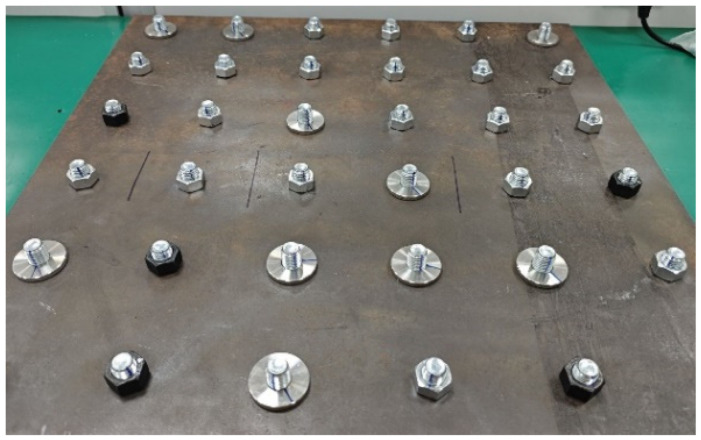
Sample intercepted images.

**Figure 12 sensors-24-06747-f012:**
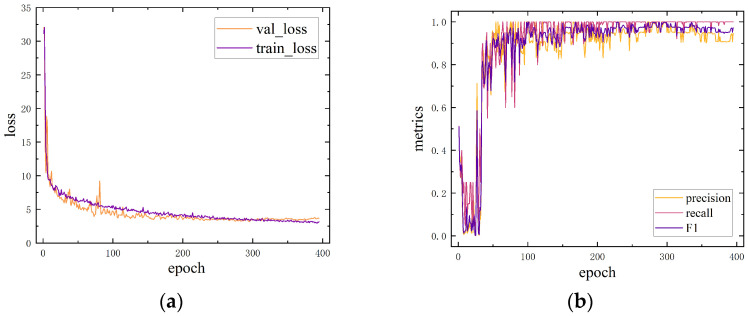
YOLOv10-S training process: (**a**) train loss and valid loss during training; (**b**) precision and recall during training.

**Figure 13 sensors-24-06747-f013:**
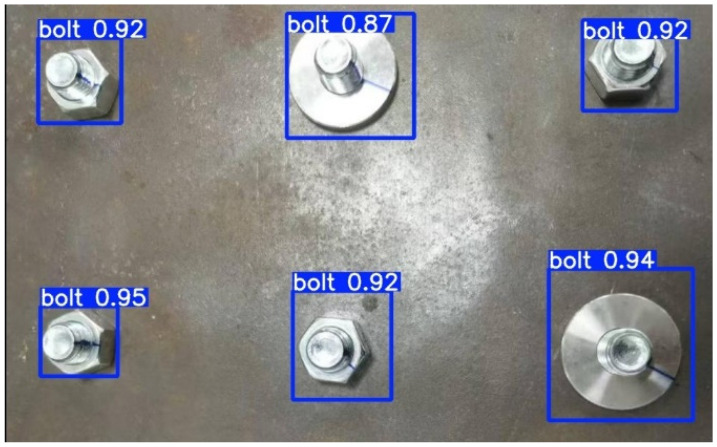
Example of YOLOv10-S detection.

**Figure 14 sensors-24-06747-f014:**
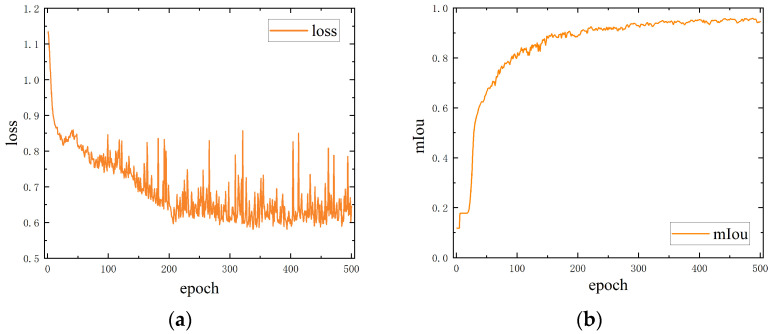
Fast-SCNN step 1 training process: (**a**) loss during step 1 training; (**b**) mIoU during step 1 training.

**Figure 15 sensors-24-06747-f015:**
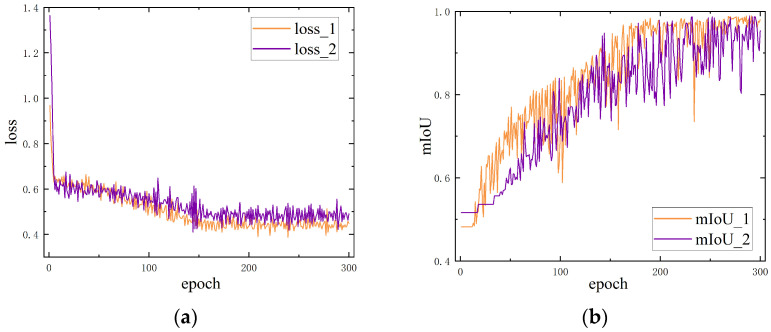
Fast-SCNN step 2 training process: (**a**) loss during step 2 training; (**b**) mIoU during step 2 training.

**Figure 16 sensors-24-06747-f016:**
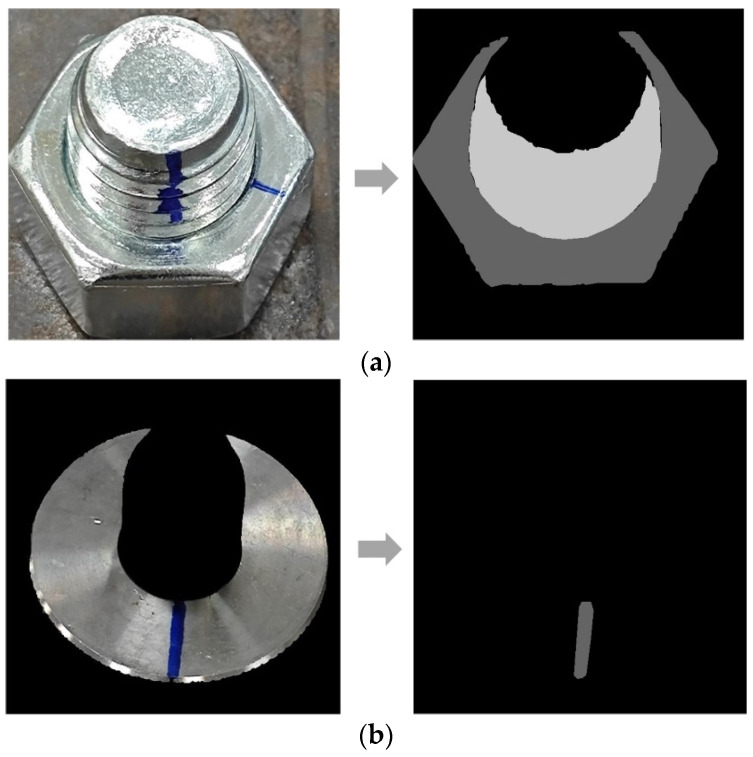
Fast-SCNN segmentation examples: (**a**) step 1 segmentation result; (**b**) step 2 segmentation result.

**Figure 17 sensors-24-06747-f017:**
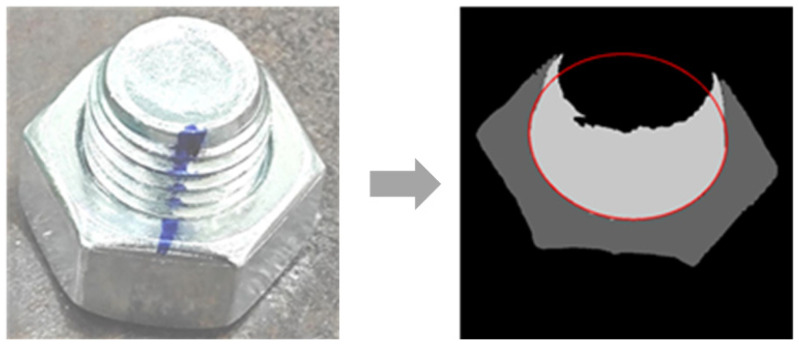
Ellipse fitting example.

**Figure 18 sensors-24-06747-f018:**
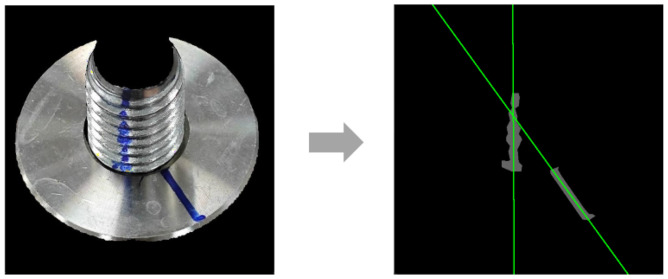
Line fitting example.

**Figure 19 sensors-24-06747-f019:**
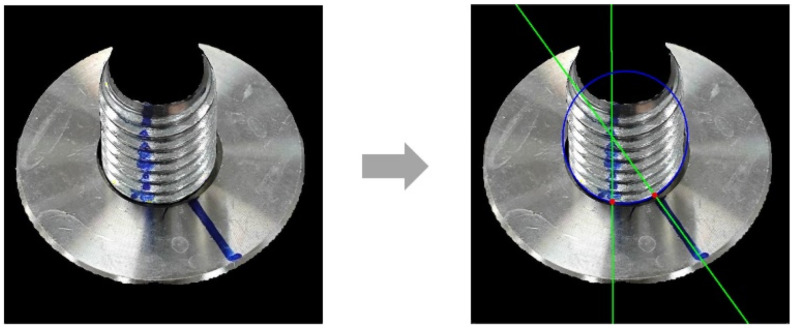
Intersection calculation examples.

**Figure 20 sensors-24-06747-f020:**
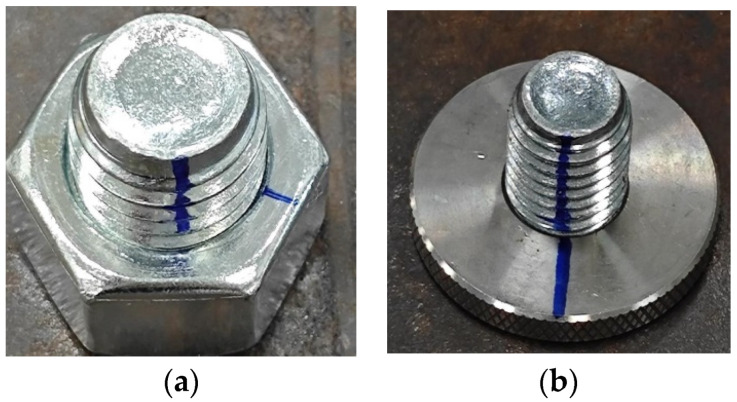
Two types of bolt markings: (**a**) bolt with hexagonal nut; (**b**) bolt with round nut.

**Figure 21 sensors-24-06747-f021:**
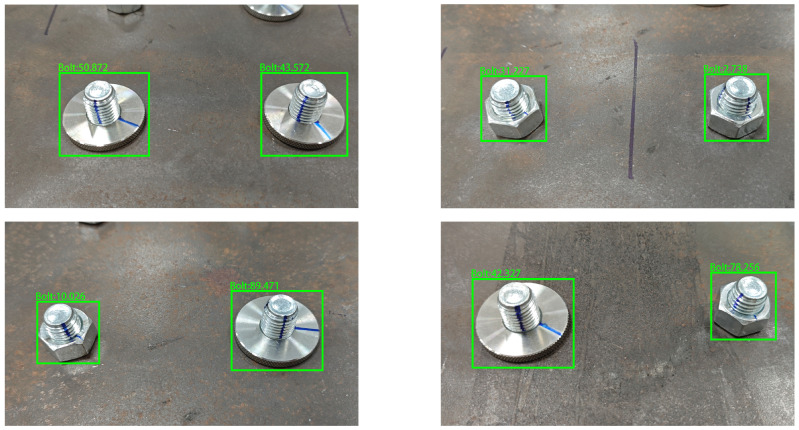
Examples of loosening angle identification.

**Table 1 sensors-24-06747-t001:** Experimental setup.

Environment Item	Configuration
CPU	Intel Core i9-13900K
GPU	NVIDIA RTX 4090
Memory	64 GB DDR5
Operating System	Ubuntu 20.04 LTS
Python Version	Python 3.11.5
CUDA Version	CUDA 12.1
PyTorch Version	PyTorch 2.1

**Table 2 sensors-24-06747-t002:** Confusion matrix.

	Predict Positive	Predict Negative
Actual Positive	True Positive (TP)	False Negative (FN)
Actual Negative	False Positive (FP)	True Negative (TN)

**Table 3 sensors-24-06747-t003:** Comparison of ellipse fitting methods.

	Error	Time Cost (ms)
OLS	65.68	**0.01**
ODR	**58.08**	0.13
LSM	87.36	15.70
ODF	404.84	633.28
RANSAC	914.52	25.59

**Table 4 sensors-24-06747-t004:** Comparison of line fitting methods.

	Error	Time Cost (ms)
OLS	7.367	0.299
TLS	**6.703**	**0.179**
Theil–Sen	7.380	22.525
Ransac	7.367	1.517

**Table 5 sensors-24-06747-t005:** Loosening angle calculation results.

	Loosening Angle	Sample Quantity	Average Calculated Loosening Angle	Average Error
0	0°	30	0.984°	0.984°
1	5°	20	4.732°	0.368
2	10°	15	11.013°	1.013°
3	30°	20	31.242°	1.242°
4	60°	15	58.845°	1.155°
5	90°	15	90.677°	0.677°
6	120°	15	121.351°	1.351°
7	150°	15	152.736°	2.736°

**Table 6 sensors-24-06747-t006:** Comparison of errors of two types of bolts.

	Bolt Type	Sample Quantity	Average Error
0	round bolt	49	0.582°
1	hexagonal bolt	96	1.432°

## Data Availability

The data cannot be made publicly available upon publication because they contain commercially sensitive information. The data that support the findings of this study are available upon reasonable request from the authors.

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
