# Peer review of "Vision-Based Real-Time Bolt Loosening Detection by Identifying Anti-Loosening Lines"

_sensors, 2024, doi:10.3390/s24206747_

Round 1
Reviewer 1 Report
Comments and Suggestions for Authors
The research paper has certain research significance, but there are the following issues:
1: Supplementary analysis of the main sources of experimental errors;
2: Conduct robustness and generalization experiments to supplement the methods used;
3: How to ensure the rationality of the experimental dataset;
Author Response
Comment 1: Supplementary analysis of the main sources of experimental errors;
Response 1: We identified the main sources of error in angle estimation as variations in image acquisition conditions, including camera angle, distance, and lighting, which affect the clarity and visibility of the anti-loosening marks. Additionally, uncertainty arises from measuring the actual angle between the marks due to line width and minor irregularities in their drawing. The image segmentation algorithm (Fast-SCNN) may introduce pixel-level errors during mark extraction, impacting the accuracy of subsequent line fitting. Although we employed the Total Least Squares (TLS) method for line fitting, it remains sensitive to noisy data. Finally, surface irregularities on the bolt can distort the captured ellipse, affecting the accuracy of ellipse fitting. All these factors contribute to the discrepancies between the algorithm's calculated loosening angle and the measured angle.
Comment 2: Conduct robustness and generalization experiments to supplement the methods used;
Response 2: We introduced various bolt types, including round nuts and hexagonal nuts, along with different loosening angles (5°, 10°, 30°, 60°, and 90°), to assess the model's performance under diverse conditions, thereby validating its adaptability and effectiveness in real-world scenarios. The results demonstrate that the model maintains high detection accuracy under different lighting conditions, confirming its robustness. However, we observed varying error rates across different bolt types and analyzed the reasons for these discrepancies. Due to commercial project requirements, the equipment used in previous experiments (camera, light source, and bolts) is no longer available in the lab, which limits further robustness testing. We plan to consider additional robustness validations once we have access to these devices again.
Comment 3: How to ensure the rationality of the experimental dataset;
Response 3:
The dataset in this study comprises 36 bolt connections, including 12 round nuts and 24 hexagonal nuts, with loosening angles set at 5°, 10°, 30°, 60°, and 90° to cover various practical conditions. The images were captured under different lighting conditions and camera angles (30° to 50°) to simulate complex field environments. To ensure representativeness, we selected common industrial bolt types and utilized a high-resolution camera (2048x1024 pixels) to guarantee clarity and visibility of the markings. The dataset was divided into 80% for training and 20% for validation, avoiding class imbalance in the training dataset. Evaluations of precision, recall, and detection speed indicated consistent model performance across different datasets, validating the dataset's rationality and the reliability of the experimental results.
Reviewer 2 Report
Comments and Suggestions for Authors
The authors in this study introduce a generalized real-time vision-based method for detecting bolt looseness by detecting anti-loosening marks on the bolt using YOLO V10s, the latest deep learning technique in object detection-additionally, a two-step Fast-SCNN is employed for image segmentation. The paper is well-written and provides sufficient analysis and visualization of experimental activities to justify the proposed method. Here are a few comments/suggestions.
1. Are the angle detection error and detection speed achieved in this study comparatively better than other related work? The abstract does not clarify this concern.
2. Generally, bolt loosening detection for example in bridges, is more effective with vibration detection via structural health monitoring, please justify, how efficient is your proposed method.
3. Is YOLOV10 architectural explanation in detail is related to your method section? Better to include the proposed method in detail.
4. The data acquisition technique is well explained and clear enough, but how the model was trained, tested, and validated with how much of the dataset? How many images were collected for training/testing/validation? Are these datasets publicly available? Please clearly provide the dataset training, testing, and validated approaches.
5. What about the harmonic mean calculation for the performance evaluation of your model, please clarify, if the harmonic mean (F1-measure) is not necessary to analyze.
Comments on the Quality of English LanguageA minor editing is suggested.
Author Response
Comment 1: Are the angle detection error and detection speed achieved in this study comparatively better than other related work? The abstract does not clarify this concern.
Response 1: We have clarified this issue in the abstract. Our proposed method demonstrates superior performance in both angle detection error and detection speed. Although direct comparisons are challenging due to inconsistencies in bolt types and detection conditions across different studies, we found that typical detection errors for similar methods range from 1° to 3°. In terms of speed, our method achieves real-time performance at 33 FPS, significantly better than 8 FPS reported in “Near Real-Time Bolt-Loosening Detection Using Mask and Region-Based Convolutional Neural Network” and 1.1 seconds in “A Fast Bolt-Loosening Detection Method of Running Train's Key Components Based on Binocular Vision.”
Comment 2: Generally, bolt loosening detection for example in bridges, is more effective with vibration detection via structural health monitoring, please justify, how efficient is your proposed method.
Response 2: Our proposed method offers significant advantages in detection efficiency: (1) It employs non-contact detection, utilizing anti-loosening line markers in high-resolution images to assess bolt looseness, avoiding complex signal processing and sensor deployment. This results in enhanced real-time performance and flexibility, particularly suited for large structures like bridges. (2) The method directly calculates the bolt loosening angle using ellipse and line fitting algorithms, achieving an average detection error of 1.145°, making it more efficient and precise compared to vibration monitoring methods, especially in scenarios requiring high accuracy. (3) Our method operates at 32 FPS on 2048x1024 pixel high-resolution images, enabling real-time monitoring and reducing response latency for quick feedback applications. (4) Additionally, the method is adaptable to various bolt types, sizes, and installation positions, providing greater flexibility and applicability compared to vibration detection methods that require complex sensor arrangements.
Comment 3: Is YOLOV10 architectural explanation in detail is related to your method section? Better to include the proposed method in detail.
Response 3: We have added a section in the paper introducing the YOLOv10 architecture. YOLOv10 is the deep learning algorithm we utilized for bolt detection, and we have provided a brief overview of its advanced features. Since our focus was primarily on the overall design of the detection method, we previously did not elaborate on YOLOv10 in detail.
Comment 4: The data acquisition technique is well explained and clear enough, but how the model was trained, tested, and validated with how much of the dataset? How many images were collected for training/testing/validation? Are these datasets publicly available? Please clearly provide the dataset training, testing, and validated approaches.
Response 4: We collected a total of 80 high-resolution images (2048x1024 pixels) for training and validation (a separate test set was not created due to the relatively small dataset). The dataset was divided into 80% for training and 20% for validation, with 64 images used for training and 16 images for validating YOLOv10 and Fast-SCNN. There is no class imbalance among the bolts and anti-loosening markers in these images. These data are part of a commercial project and, due to confidentiality agreements, are not publicly available at this time.
Comment 5: What about the harmonic mean calculation for the performance evaluation of your model, please clarify, if the harmonic mean (F1-measure) is not necessary to analyze.
Response 5: We have now included the F1 score as a performance metric. Initially, we focused only on precision and recall for model evaluation, omitting the F1 score. This choice was made because precision and recall directly reflect the model's performance in bolt loosening detection tasks. Precision measures the proportion of true positive samples among predicted positives, while recall reflects the proportion of actual positives correctly identified. Recognizing the importance of the F1 score as a combined metric of precision and recall, we have now added the F1 score curve to our results.
Reviewer 3 Report
Comments and Suggestions for Authors
The paper proposes a method employs the YOLOv10-S deep learning model for high-precision, real-time bolt detection, followed by a two-step Fast-SCNN image segmentation process. I recommended major revision.
1. Please highlight your contribution.
2.Literature survey is not well organized, missing many works about multimedia processing.
For example,
TSD-CAM: transformer-based self distillation with CAM similarity for weakly supervised semantic segmentation
An Intelligent Weighted Object Detector for Feature Extraction to Enrich Global Image Information
Disentangled Dynamic Deviation Transformer Networks for Multivariate Time Series Anomaly Detection
3.Figures are low quality, please improve the appearance of figures.
4.The authors should compare with the state-of-the-art. Besides, analysis for experiment results is not sufficient.
specific comments:
This article proposes a real-time bolt loosening detection method based on vision, which achieves real-time monitoring of bolt status by identifying anti loosening lines. This method has important application value in the field of structural health monitoring and has a certain degree of innovation. However, during the review process, I also identified some areas that need improvement.
1.The author should further elaborate on the shortcomings of existing bolt loosening detection methods, as well as how this study overcomes these shortcomings, in order to more clearly demonstrate the necessity and importance of this research.
2.Literature survey is not well organized, missing many works about multimedia processing.
For example,
TSD-CAM: transformer-based self distillation with CAM similarity for weakly supervised semantic segmentation
An Intelligent Weighted Object Detector for Feature Extraction to Enrich Global Image Information
Disentangled Dynamic Deviation Transformer Networks for Multivariate Time Series Anomaly Detection
3.Although the author provides a detailed description of the method for identifying anti loosening lines, some technical details are still not clear enough. Suggest the author to add algorithm descriptions, flowcharts, or pseudocode to help readers better understand and reproduce the experimental results. In addition, the author should discuss the possible technical challenges and limitations of this method and propose corresponding solutions.
4.The experimental design is basically reasonable, but the analysis of the experimental results is slightly simple. Suggest the author to add more experimental data and conduct more in-depth data analysis to verify the reliability and effectiveness of the method. At the same time, the author should discuss possible sources of error in the experimental results and provide suggestions for improving the experimental design.
5. The overall structure of the paper is clear, but the language expression in some paragraphs is slightly lengthy. Suggest the author to optimize language expression to make the paper more concise and clear. In addition, the format of some charts needs to be standardized to ensure the academic rigor of the paper.
6. This study has a certain degree of innovation in the field of bolt loosening detection, but the author should further emphasize its innovative points and contributions, such as the proposed new methods and solved new problems. Meanwhile, the author can compare with other related studies to highlight the uniqueness and advantages of this research.
Comments on the Quality of English Language
The text requires some revision for syntactical and grammatical mistakes.
Author Response
Comments 1: Please highlight your contribution.
Response 1: We have enhanced the description of our contributions in the introduction and conclusion sections. The main contributions of this study are as follows: (1) We propose a real-time bolt loosening detection method based on computer vision, integrating the YOLOv10-S deep learning model with Fast-SCNN image segmentation techniques. This method enables the direct identification of anti-loosening line markers at the bolt connection without relying on the historical state of the bolts. (2) Our method achieves an average angle detection error of 1.145° and a detection speed of 32 FPS under high-resolution images (2048x1024 pixels), demonstrating superior balance in detection accuracy and speed compared to traditional and other vision-based methods. (3) The method shows excellent adaptability to different types of bolts and is applicable to various complex monitoring scenarios, showcasing significant practical application potential.
Comments 2:
Literature survey is not well organized, missing many works about multimedia processing.
For example,
TSD-CAM: transformer-based self distillation with CAM similarity for weakly supervised semantic segmentation
An Intelligent Weighted Object Detector for Feature Extraction to Enrich Global Image Information
Disentangled Dynamic Deviation Transformer Networks for Multivariate Time Series Anomaly Detection
Response 2:
We have supplemented the literature review to include relevant studies in the field. However, this research primarily focuses on bolt loosening detection using computer vision and deep learning methods. Therefore, the review emphasizes literature related to bolt loosening detection rather than multimedia processing algorithms. We acknowledge advanced multimedia techniques such as TSD-CAM, intelligent weighted object detectors, and disentangled dynamic deviation transformer networks; however, given the specific focus of our study, these techniques do not directly address the core issues of bolt loosening detection. Thus, we concentrated on analyzing recent research outcomes related to structural health monitoring and applications in computer vision relevant to bolt loosening detection.
Comments 3: Figures are low quality, please improve the appearance of figures.
Response 3:
In response to your comment regarding the low quality of graphics, we have replaced unclear images in the manuscript to ensure that all figures meet the required resolution and detail standards for publication.
Comments 4: The authors should compare with the state-of-the-art. Besides, analysis for experiment results is not sufficient.
Response 4: We have included a review of recent research findings and conducted a more thorough analysis of the experimental results. Due to differences in experimental environments, bolt types, and result standards in the field of bolt loosening detection, direct comparisons of results can be challenging. However, we have made efforts to provide a more detailed analysis of our method's performance based on the experimental results.
specific comments:
Comments 1: The author should further elaborate on the shortcomings of existing bolt loosening detection methods, as well as how this study overcomes these shortcomings, in order to more clearly demonstrate the necessity and importance of this research.
Response 1: Currently, bolt loosening detection primarily relies on traditional methods, including vibration analysis, torque measurement, and contact sensors. These methods have notable limitations: (1) reliance on sensor placement increases system complexity and cost; (2) environmental sensitivity can lead to unstable detection results; (3) limited precision in complex scenarios makes it difficult to meet safety requirements; and (4) inadequate real-time monitoring capabilities. To address these issues, we propose a real-time bolt loosening detection method based on computer vision, offering non-contact detection, strong robustness, high accuracy (1.145°), and adaptability to various bolt types. This solution significantly improves the reliability and efficiency of detection.
Comments 2:
Literature survey is not well organized, missing many works about multimedia processing.
For example,
TSD-CAM: transformer-based self distillation with CAM similarity for weakly supervised semantic segmentation
An Intelligent Weighted Object Detector for Feature Extraction to Enrich Global Image Information
Disentangled Dynamic Deviation Transformer Networks for Multivariate Time Series Anomaly Detection
Response 2: As previously mentioned, we have enhanced our literature review to ensure it is systematic and comprehensive while remaining focused on bolt loosening detection.
Comments 3: Although the author provides a detailed description of the method for identifying anti- loosening lines, some technical details are still not clear enough. Suggest the author to add algorithm descriptions, flowcharts, or pseudocode to help readers better understand and reproduce the experimental results. In addition, the author should discuss the possible technical challenges and limitations of this method and propose corresponding solutions.
Response 3: We have added more detailed algorithm descriptions and relevant technical details to clarify the key steps in the anti-loosening line identification method. Additionally, we provided a flowchart to illustrate the entire detection process, aiding readers in understanding the algorithm’s logic and implementation steps.
Comments 4: The experimental design is basically reasonable, but the analysis of the experimental results is slightly simple. Suggest the author to add more experimental data and conduct more in-depth data analysis to verify the reliability and effectiveness of the method. At the same time, the author should discuss possible sources of error in the experimental results and provide suggestions for improving the experimental design.
Response 4: We have increased the depth of our experimental analysis and discussion to support the reliability and effectiveness of the method. Additionally, we examined potential sources of error in the experimental results, including measurement errors, image segmentation errors, line fitting errors, and discrepancies in marking accuracy.
Comments 5: The overall structure of the paper is clear, but the language expression in some paragraphs is slightly lengthy. Suggest the author to optimize language expression to make the paper more concise and clear. In addition, the format of some charts needs to be standardized to ensure the academic rigor of the paper.
Response 5: We have conducted a comprehensive language optimization to ensure that the writing is more concise and clear. We also standardized the formatting of figures to enhance the overall academic rigor of the paper.
Comments 6: This study has a certain degree of innovation in the field of bolt loosening detection, but the author should further emphasize its innovative points and contributions, such as the proposed new methods and solved new problems. Meanwhile, the author can compare with other related studies to highlight the uniqueness and advantages of this research.
Response 6: We have clearly articulated the innovative aspects of this study in both the introduction and conclusion, highlighting the unique contributions of our real-time bolt loosening detection method, including its non-contact capability and superior performance in terms of adaptability and accuracy compared to traditional methods.
Round 2
Reviewer 1 Report
Comments and Suggestions for Authors The manuscript has been revised according to the reviewers' comments, and the innovation meets the requirements. It is recommended for acceptance.